



**Landslide susceptibility assessment of the part of the North Anatolian Fault Zone**
**(Turkey) by GIS-based frequency ratio and index of entropy models**

4                                    Gökhan Demir[1]
[1]Department of Civil Engineering, Faculty of Engineering, Ondokuz Mayıs University,
7                                    Samsun, Turkey
8                          Correspondence: gokhan.demir@omu.edu.tr

**Abstract:** In the present study, landslide susceptibility assessment for the the part of the
North Anatolian Fault Zone is made using index of entropy models within geographical
information system. At first, the landslide inventory map was prepared in the study area
using earlier reports, aerial photographs and multiple field surveys. 63 cases (69 %) out
of 91 detected landslides were randomly selected for modeling, and the remaining 28
(31 %) cases were used for the model validation. The landslide-trigerring factors,
including slope degree, aspect, elevation, distance to faults, distance to streams, distance
to road. Subsequently, landslide susceptibility maps were produced using frequency
ratio and index of entropy models. For verification, the receiver operating characteristic
(ROC) curves were drawn and the areas under the curve (AUC) calculated. The
verification results showed that frequency ratio model (AUC=75.71%) performed
slightly better than index of entropy (AUC=75.43%) model. The interpretation of the
susceptibility map indicated that distance to streams, distance to road and slope degree
play major roles in landslide occurrence and distribution in the study area. The landslide
susceptibility maps produced from this study could assist planners and engineers for
reorganizing and planning of future road construction.
**Keywords:** Landslide susceptibility, GIS, Nort Anatolian Fault Zone, İndex of Entropy,
Reşadiye, Tokat.

30       **1. Introduction**
Among various natural hazards, landslides are the most widespread and damaging.
Potentials landslide-prone areas should, therefore, are identified in advance in order to
reduce such damage. In this respect, landslide susceptibility assessment can provide
valuable information essential for hazard mitigation through proper project planning
and implementation. The main goal of landslide susceptibility analysis is to identify
dangerous and high risk areas and thus landslide damage can be reduced through
suitable mitigation measures ( Solaimani *et al.* 2013). Different methods to prepare
landslide susceptibility and hazard maps using statistical methods and GIS tools were
developed in the last decade (Van Westen et al. 2003; Guzzetti et al. 2005). Many of
these studies have applied statistical models such as logistic regression(Akgun 2012;
Ozdemir and Altural 2013; Solaimani et al. 2013;Demir et al. 2015), bivariate(Bednarik
et al. 2010; Pareek et al. 2010; Pradhan and Youssef 2010)  and multivariate (Pradhan
2010a, b; Choi et al. 2012). Probabilistic models such as frequency ratio (FR), weight of
evidence (WOE), etc. have been used in landslide susceptibility mapping (Akgun et al.
2008; Oh et al. 2012; Yilmaz and Keskin 2009; Youssef et al. 2009, 2013; Pradhan and
Youssef 2010; Pradhan et al. 2011; Akgun et al 2012; Saponaro et al. 2014; Sujatha et
al. 2014; Demir et al. 2015, Bourenane et al 2016, Chen et al. 2016b). Other different
methods such as, analytical hierarchy process (AHP) (Yalcin et al. 2011; Pourghasemi





et al. 2012a; Park et al. 2013, Demir et al. 2013, Myronidis et al. 2016, Wu et al. 2016,
Chen et al. 2016a), index of entropy (IOE) model(Mihaela et al.2011; Devkota et al.
2013, Jaafari et al 2013, Youssef et al. 2015, Wang et al. 2016), certainty factor (CF) model
(Devkota et al. 2013), artificial neural network model (Chauhan et al. 2010; Pouydal et
al. 2010; Pradhan and Buchroithner 2010; Park et al. 2013),, spatial multicriteria
decision analysis (MCDA) approach (Akgun and Turk 2010; Akgun 2012), fuzzy logic
and neuro-fuzzy (Vahidnia et al. 2010; Sezer et al. 2011), decision-tree methods
(Nefeslioglu et al. 2010; Pradhan 2013), fuzzy logic (Pourghasemi et al. 2012a, 2012b,
2012c; Sharma et al. 2013), support vector machine (SVM) (Yilmaz 2010; Pradhan
2013) have also been employed for the purpose of landslide susceptibility mapping.
This study aims to develop landslide susceptibility maps of the part of the North
Anatolian Fault Zone, southeast Resadiye-Koyulhisar Turkey, (Fig. 1), using index of
entropy (IOE) model. To achieve this, index of entropy analysis methodology, to obtain
landslide susceptibility map using the geographic information system was developed,
applied, and verified in the study area.

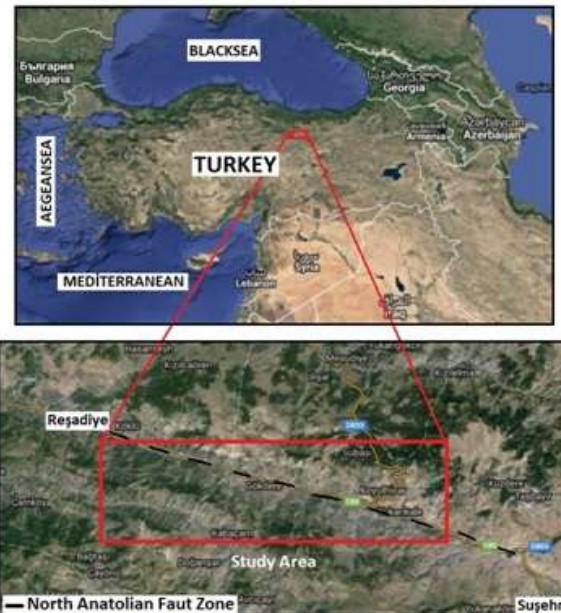


**Figure 1.** Study Area

**2. Study area**
The study area is located in the the North Anatolian Fault Zone, between the southeast
Resadiye to Koyulhisar Sivas province.   The area lies between $44^{o}70'64''$ and $44^{o}$
$56'84$ latitude and $36^{o} 21'87$ and $41^{o} 47'09''$ longitude, and covers an area about of
$720 \text{ km}^2$.
According to a geological map prepared by General Directory of Mineral Research and
Exploration, north of the NAFZ there are, from old to young, Upper Cretaceous-age
volcanic and sedimentary units, Maastrichtian-age limestone, and Pliocene-age basalt





and other volcanic units. While the Upper Cretaceous volcanic and sedimentary units on
the lower slopes have a gentle slope morphology, Maastrichtian limestones present a
very steep morphology. While the dip of the lower beds of the limestone varies over
short distances due to the effect of the NAFZ, it is generally to the northeast (Gokceoglu
et al. 2005). Landslides are common natural hazards in the seismically active North
Anatolian Fault Zone (NAFZ), which is 1,100 km in length and is moving westward
with the rate of 2.5 cm every year according to geological and GPS data(Demir et al
2013) The latest catastrophic event occurred on March 17, 2005 at Kuzulu (Sivas) in the
valley. The landslide was initiated within highly weathered volcanics in the mode of
sliding and then transformed to an earth flow. It killed 15 people, and more than 30
houses and a mosque were buried and damaged by the earth flow material. A second but
smaller landslide originated from the same source areas after 4 days and caused
additional damages (Gokceoglu et al. 2005; Ulusay et al. 2007; Yılmaz 2009). After the
main event, the governor needed landslide susceptibility maps of the landslide area.
**3. Data production**
The study began with the preparation of a landslide inventory map based on field work,
earlier reports and satellite images. Landslide inventory maps show the areal
distribution of existing landslide areas and their characteristics. These maps indicate the
landslides, which are perceptible on site (Cevik and Topal 2003). In total, 321
landslides were mapped (Fig. 2) and subsequently digitized for further analysis. The
mapped landslides cover an area of 47.81 km2, which constitutes 6.68 % of the entire
study area. From these landslides, 63 (69 %) randomly selected instabilities were taken
for making landslide susceptibility models and 28 (31 %) were used for validating the
models.

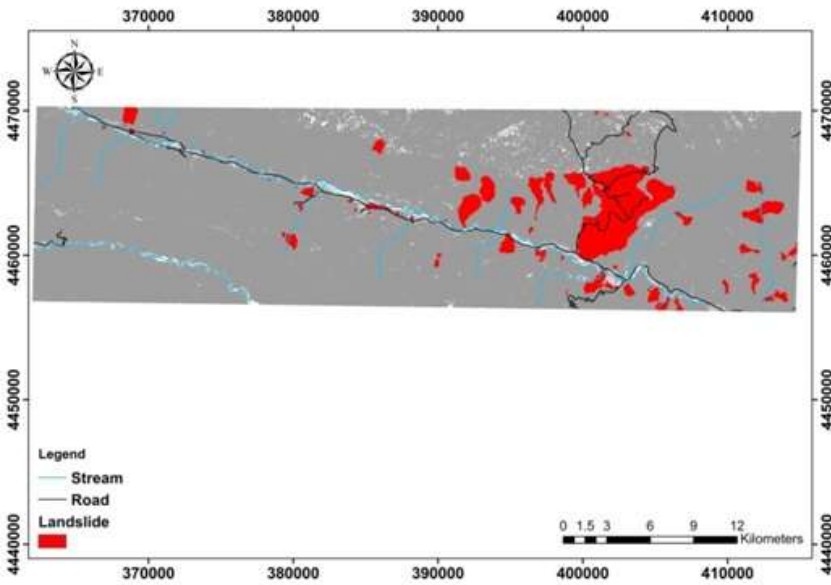

**Figure 2.** Landslide inventory map




The number of landslide-conditioning factors may range from only a few numbers to
several (Mohammady et al. 2012; Pourghasemi et al. 2012d; Papathanassiou et al.
2012). The selection of these factors mainly depends on the availability of data for the
study area and the relevance with respect to landslide occurrences (Papathanassiou et
al.2012). According to the Ayalew and Yamagishi (2005), in GIS-based studies, the
selected factors should be operational, complete, non-uniform, measurable, and non-
redundant. We prepared six thematic data layers representing the following landslide
conditioning factors: slope degree, aspect, elevation, distance to faults, distance to
streams, distance to road (Figure 3).

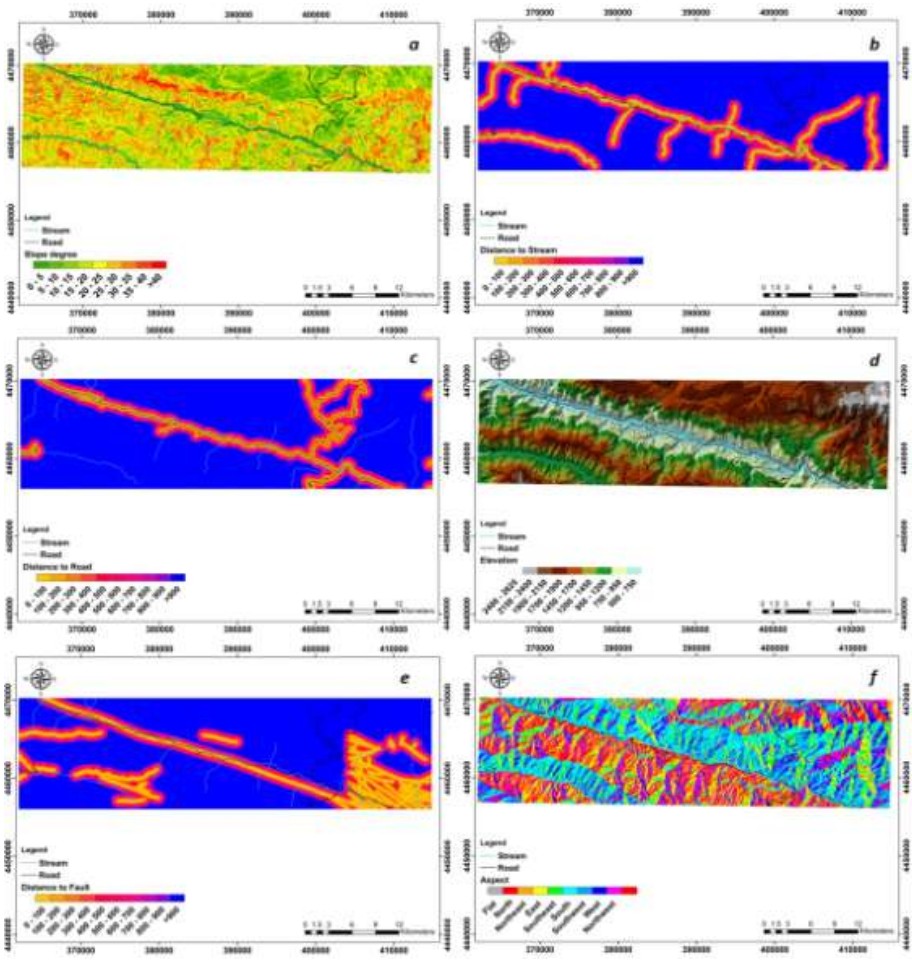


**Figure 3.** Conditioning Factors(a.slope degree, b.distance to stream, c.distance to road,
d.elevation, e.distance to faults,f.aspect.)



The main parameter of the slope stability analysis is the slope degree(S. Lee and K. Min
2001). Because the slope degree is directly related to the landslides, it is frequently used
in preparing landslide susceptibility maps[S. Lee, J. H. Ryu, J. S. Won and H. J. Park
(2004), M. Ercanoglu, C. Gokceoglu, T. W. J. Van Asch (2004, A. Clerici, S. Perego, C.
Tellini and P. Vescovi (2002) . For this reason, the slope degree map of the study area is
prepared from the digital elevation model (DEM) and divided into nine slope classes
with an interval of 5°(Figure 3). Aspect and elevation also were extracted from the
DEM. Aspects are grouped into 9 classes such as flat (-1), north (337.5°–360°, 0°–
22.5°), northeast (22.5°–67.5°), east (67.5°–112.5°), southeast (112.5°–157.5°), south
(157.5°–202.5°), southwest (202.5°–247.5°), west (247.5°–292.5°), and northwest
(292.5°–337.5°).In the study area, the elevation ranges between 500 and 2,625 m. The
elevation values were divided into nine classes (Figure 3). The distance from faults,
road and stream is calculated at 100m intervals using the geological map (Figure 9). An
important parameter that controls the stability of a slope is the saturation degree of the
material on the slope(Yalçın 2008, Yalçın and Bulut 2007). The closeness of the slope
to drainage structures is another important factor in terms of stability. Streams may
adversely affect stability by eroding the slopes or by saturating the lower part of
material resulting in water level increases(Pourghasemi et al. 2012a, Gökçeoğlu 1996,
Saha et al. 2002). For this reason, ten different buffer zones were created within the
study area to determine the degree to which the streams affected the slopes. A road
constructed can cause a disturbance of the slopes that lead to increase in stress on the
back of the slope, because of changes in topography and decrease of load on toe, some
tension cracks may develop. Although a slope is balanced before the road construction,
some instability may be happened because of negative effects of excavation. In the
current study many landslides were recorded along the roads in the study area that is
due to road construction. The distance from roads was calculated and reclassified into
ten classes.
**4. Landslide Susceptibility Analysis**
**a. Application of Index of Entropy Model**
In this study index of entropy model was used for landslide susceptibility analysis using
six landslide conditioning factors.
The entropy indicates the extent of the instability, disorder, imbalance, and uncertainty
of a system (Yufeng and Fengxiang, 2009). The entropy of a landslide refers to the
extent that various factors influence the development of a landslide (Pourghasemi et al.,
2012b; Jaafari et al., 2013). Several important factors provide additional entropy into
the index system. Therefore, the entropy value can be used to calculate objective
weights of the index system. The equations used to calculate the information coefficient
Wj representing the weight value for the parameter as a whole (Bednarik et al., 2010;
Constantin et al., 2011) are given as follows:





$$P_{ij} = \frac{b}{a} \tag{1}$$

$$(P_{ij}) = \frac{P_{ij}}{\sum\limits_{j=1}^{S_j} P_{ij}} \tag{2}$$

$$H_j = -\sum\limits_{i=1}^{S_j} (P_{ij}) \log_2 (p_{ij}), \; j = 1,..,n \tag{3}$$

$$H_{j\,max} = \log_2 S_j \; S_j - number\ of\ classes \tag{4}$$

$$I_j = \frac{H_{j\,max} - H_j}{H_{j\,max}}, \; I = (0,1), \; j = 1,...,n \tag{5}$$

$$W_j = I_j.P_{ij} \tag{6}$$

where a and b are the domain and landslide percentages, respectively, $(P_{ij})$ is the probability density, $H_j$ and $H_j$ max represent entropy values, $I_j$ is the information coefficient and $W_j$ represents the resultant weight value for the parameter as a whole.
The final landslide susceptibility map was prepared by the summation of weighted products of the secondarily parametric maps. The final landslide susceptibility maps using index of entropy model was developed using the following equation:

$$Y_{iIE} = ((Slope\ \text{deg}\ ree * 0,110) + (Aspect * 0,080) +$$
$$(Elevation * 0,136) + (Dis\tan ce\ to\ Road * 0,023) +$$
$$(Dis\tan ce\ to\ Stream * 0,023) + (Dis\tan ce\ to\ Fault) * 0,005)) \tag{7}$$

where Y is the value of landslide susceptibility (Fig. 4). The result of this summation is a continuous interval of values from 0.7297 to 6.1861, which represents the landslide susceptibility index. A natural break classification method was used to divide the interval into four classes and a susceptibility map was prepared (Bednarik et al., 2010; Constantin et al., 2011; Erner et al., 2010; Falaschi et al., 2009; Ram Mohan et al., 2011; Xu et al., 2012a, 2012b). According to the landslide susceptibility map generated with the IOE model (Fig. 4 and Table 1), it was found that 24.87% and 23.50% of the total landslides falls in the very low and low susceptibility zones respectively. Moderate, high, and very high susceptible zones represent 20.37%, 16.42%, and 14.83% of the landslides pixels, respectively.

**Table.1** Spatial relationship between each landslide conditioning factor and landslides using index of entropy model.

| Factor (Parameter) | Class | Percentage of pixels in the class (%) (a) | Percentage of landslide pixels(%) (b) | Pij | (Pij) | Hj | Hjmax | Ij | Wj |
|---|---|---|---|---|---|---|---|---|---|
| Elevation (m) | 500-750 | 11.324 | 10.406 | 0.919 | 0.147 | 2.551 | 3.170 | 0,195 | 0,136 |
| | 750-950 | 13.048 | 16.226 | 1.244 | 0.199 | | | | |
| | 950-1200 | 18.265 | 21.566 | 1.181 | 0.189 | | | | |
| | 1200-1450 | 18.478 | 24.770 | 1.341 | 0.214 | | | | |
| | 1450-1700 | 19.762 | 20.921 | 1.059 | 0.169 | | | | |



| | | | | | | | | | |
|---|---|---|---|---|---|---|---|---|---|
| | 1700-1900 | 12.213 | 6.021 | 0.493 | 0.079 | | | | |
| | 1900-2150 | 4.096 | 0.091 | 0.022 | 0.004 | | | | |
| | 2150-2400 | 2.115 | 0.000 | 0.000 | 0.000 | | | | |
| | 2400-2625 | 0.699 | 0.000 | 0.000 | 0.000 | | | | |
| Slope degree (°) | 0-5 | 7.093 | 7.888 | 1.112 | 0.142 | 2.770 | 3.170 | 0.126 | 0,110 |
| | 5-10 | 11.692 | 24.212 | 2.071 | 0.265 | | | | |
| | 10-15 | 15.430 | 26.660 | 1.728 | 0.221 | | | | |
| | 15-20 | 16.929 | 17.950 | 1.060 | 0.136 | | | | |
| | 20-25 | 16.714 | 11.004 | 0.658 | 0.084 | | | | |
| | 25-30 | 15.076 | 6.438 | 0.427 | 0.055 | | | | |
| | 30-35 | 10.864 | 4.399 | 0.405 | 0.052 | | | | |
| | 35-40 | 4.870 | 1.338 | 0.275 | 0.035 | | | | |
| | >40 | 1.331 | 0.111 | 0.083 | 0.011 | | | | |
| Aspect | FLAT | 1.043 | 0.302 | 0.290 | 0.034 | 2.871 | 3.170 | 0,094 | 0,080 |
| | NORTH | 13.922 | 5.270 | 0.378 | 0.045 | | | | |
| | NORTHEAST | 12.046 | 4.593 | 0.381 | 0.045 | | | | |
| | EAST | 9.461 | 5.779 | 0.611 | 0.072 | | | | |
| | SOUTHEAST | 10.956 | 11.476 | 1.047 | 0.124 | | | | |
| | SOUTH | 15.267 | 25.409 | 1.664 | 0.196 | | | | |
| | SOUTHWEST | 13.818 | 24.351 | 1.762 | 0.208 | | | | |
| | WEST | 11.493 | 15.909 | 1.384 | 0.163 | | | | |
| | NORTHWEST | 11.994 | 6.911 | 0.576 | 0.068 | | | | |
| Distance to Stream (m) | 0-100 | 3.692 | 7.789 | 2.110 | 0.152 | 3.266 | 3.322 | 0,017 | 0,023 |
| | 100-200 | 3.630 | 7.086 | 1.952 | 0.140 | | | | |
| | 200-300 | 3.584 | 5.757 | 1.606 | 0.115 | | | | |
| | 300-400 | 3.614 | 5.274 | 1.459 | 0.105 | | | | |
| | 400-500 | 3.522 | 4.877 | 1.384 | 0.100 | | | | |
| | 500-600 | 3.440 | 4.510 | 1.311 | 0.094 | | | | |
| | 600-700 | 3.307 | 3.666 | 1.108 | 0.080 | | | | |
| | 700-800 | 3.133 | 3.451 | 1.101 | 0.079 | | | | |
| | 800-900 | 3.110 | 3.400 | 1.093 | 0.079 | | | | |
| | >900 | 68.968 | 54.190 | 0.786 | 0.056 | | | | |
| Distance to Road (m) | 0-100 | 6.937 | 12.505 | 1.803 | 0.142 | 3.262 | 3.322 | 0,018 | 0,023 |
| | 100-200 | 6.445 | 10.563 | 1.639 | 0.130 | | | | |
| | 200-300 | 6.258 | 9.866 | 1.577 | 0.125 | | | | |
| | 300-400 | 6.179 | 9.075 | 1.469 | 0.116 | | | | |
| | 400-500 | 5.825 | 7.848 | 1.347 | 0.106 | | | | |
| | 500-600 | 5.016 | 5.799 | 1.156 | 0.091 | | | | |
| | 600-700 | 5.813 | 6.422 | 1.105 | 0.087 | | | | |
| | 700-800 | 5.801 | 5.880 | 1.014 | 0.080 | | | | |
| | 800-900 | 6.566 | 6.414 | 0.977 | 0.077 | | | | |
| | >900 | 45.159 | 25.628 | 0.568 | 0.045 | | | | |
| Distance to | 0-100 | 6.331 | 4.788 | 0.756 | 0.079 | 3.305 | 3.322 | 0,005 | 0,005 |



| Fault (m) | | | | |
|---|---|---|---|---|
| 100-200 | 5.193 | 4.444 | 0.856 | 0.090 |
| 200-300 | 4.245 | 3.980 | 0.938 | 0.098 |
| 300-400 | 3.682 | 3.346 | 0.909 | 0.095 |
| 400-500 | 3.261 | 3.676 | 1.127 | 0.118 |
| 500-600 | 2.940 | 3.592 | 1.222 | 0.128 |
| 600-700 | 2.984 | 3.167 | 1.061 | 0.111 |
| 700-800 | 2.899 | 2.548 | 0.879 | 0.092 |
| 800-900 | 2.805 | 2.153 | 0.768 | 0.080 |
| >900 | 65.660 | 68.306 | 1.040 | 0.109 |


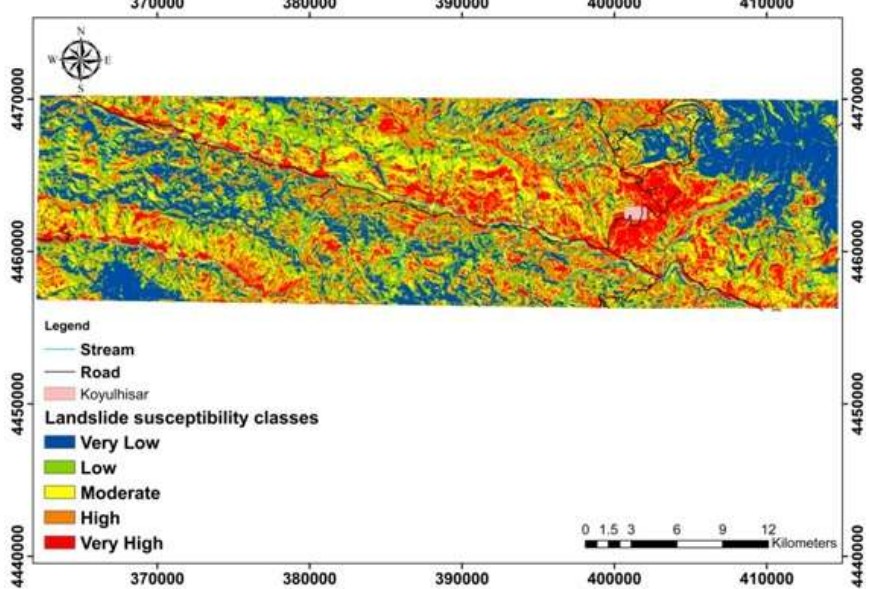

**Figure 4.** Landslide susceptibility map IOE
185             **b.  Application of Frequency ratio method**

Frequency ratio method is a simple and understandable probabilistic model, and the
model is based on the observed relationships between distribution of landslides and
each landslide-causative factor, to reveal the correlation between landslide locations and
the factors in the study area (Lee and Pradhan, 2007). To calculate the frequency ratio,
the ratio of landslide occurrence to non-occurrence (Regmi et al., 2013) was calculated
for each factor's class. Therefore, the frequency ratio for each factor's class was
calculated from its relationship with landslide events. The frequency ratio is defined as
shown in Equation (8).

$$FR = \frac{PLO}{PIF} \qquad\qquad (8)$$





Here, PLO is the subcategory percentage of each factor conditioning landslide in a landslide area, while PIF is the category percentage of each factor conditioning landslide (Table 2).

**Table.2** Frequency ratio values of the landslide-trigerring parameters.

| Factor (Parameter) | Class | Number of pixels in class | Percentage of pixels in the class (%) (a) | Number of landslide pixels | Percentage of landslide pixels(%) (b) | FREQUENCY RATIO (FR) (b/a) |
|---|---|---|---|---|---|---|
| Elevation (m) | 500-750 | 130386 | 11.324 | 5166 | 10.406 | 0.919 |
| | 750-950 | 150242 | 13.048 | 8055 | 16.226 | 1.244 |
| | 950-1200 | 210313 | 18.265 | 10706 | 21.566 | 1.181 |
| | 1200-1450 | 212763 | 18.478 | 12297 | 24.770 | 1.341 |
| | 1450-1700 | 227550 | 19.762 | 10386 | 20.921 | 1.059 |
| | 1700-1900 | 140627 | 12.213 | 2989 | 6.021 | 0.493 |
| | 1900-2150 | 47161 | 4.096 | 45 | 0.091 | 0.022 |
| | 2150-2400 | 24348 | 2.115 | 0 | 0.000 | 0.000 |
| | 2400-2625 | 8044 | 0.699 | 0 | 0.000 | 0.000 |
| Slope degree (°) | 0-5 | 81676 | 7.093 | 3916 | 7.888 | 1.112 |
| | 5-10 | 134628 | 11.692 | 12020 | 24.212 | 2.071 |
| | 10-15 | 177671 | 15.430 | 13235 | 26.660 | 1.728 |
| | 15-20 | 194923 | 16.929 | 8911 | 17.950 | 1.060 |
| | 20-25 | 192448 | 16.714 | 5463 | 11.004 | 0.658 |
| | 25-30 | 173594 | 15.076 | 3196 | 6.438 | 0.427 |
| | 30-35 | 125095 | 10.864 | 2184 | 4.399 | 0.405 |
| | 35-40 | 56076 | 4.870 | 664 | 1.338 | 0.275 |
| | >40 | 15323 | 1.331 | 55 | 0.111 | 0.083 |
| Aspect | FLAT | 12005 | 1.043 | 150 | 0.302 | 0.290 |
| | NORTH | 160305 | 13.922 | 2616 | 5.270 | 0.378 |
| | NORTHEAST | 138703 | 12.046 | 2280 | 4.593 | 0.381 |
| | EAST | 108940 | 9.461 | 2869 | 5.779 | 0.611 |
| | SOUTHEAST | 126147 | 10.956 | 5697 | 11.476 | 1.047 |
| | SOUTH | 175786 | 15.267 | 12614 | 25.409 | 1.664 |
| | SOUTHWEST | 159110 | 13.818 | 12089 | 24.351 | 1.762 |
| | WEST | 132340 | 11.493 | 7898 | 15.909 | 1.384 |
| | NORTHWEST | 138098 | 11.994 | 3431 | 6.911 | 0.576 |
| Distance to Stream (m) | 0-100 | 42513 | 3.692 | 3867 | 7.789 | 2.110 |
| | 100-200 | 41798 | 3.630 | 3518 | 7.086 | 1.952 |
| | 200-300 | 41271 | 3.584 | 2858 | 5.757 | 1.606 |
| | 300-400 | 41614 | 3.614 | 2618 | 5.274 | 1.459 |
| | 400-500 | 40558 | 3.522 | 2421 | 4.877 | 1.384 |
| | 500-600 | 39604 | 3.440 | 2239 | 4.510 | 1.311 |
| | 600-700 | 38082 | 3.307 | 1820 | 3.666 | 1.108 |
| | 700-800 | 36070 | 3.133 | 1713 | 3.451 | 1.101 |
| | 800-900 | 35807 | 3.110 | 1688 | 3.400 | 1.093 |
| | >900 | 794117 | 68.968 | 26902 | 54.190 | 0.786 |
| Distance to Road (m) | 0-100 | 79875 | 6.937 | 6208 | 12.505 | 1.803 |





| | | | | | |
|---|---|---|---|---|---|
| | 100-200 | 74209 | 6.445 | 5244 | 10.563 | 1.639 |
| | 200-300 | 72053 | 6.258 | 4898 | 9.866 | 1.577 |
| | 300-400 | 71146 | 6.179 | 4505 | 9.075 | 1.469 |
| | 400-500 | 67076 | 5.825 | 3896 | 7.848 | 1.347 |
| | 500-600 | 57759 | 5.016 | 2879 | 5.799 | 1.156 |
| | 600-700 | 66937 | 5.813 | 3188 | 6.422 | 1.105 |
| | 700-800 | 66800 | 5.801 | 2919 | 5.880 | 1.014 |
| | 800-900 | 75608 | 6.566 | 3184 | 6.414 | 0.977 |
| | >900 | 519971 | 45.159 | 12723 | 25.628 | 0.568 |
| Distance to Fault (m) | 0-100 | 72894 | 6.331 | 2377 | 4.788 | 0.756 |
| | 100-200 | 59789 | 5.193 | 2206 | 4.444 | 0.856 |
| | 200-300 | 48884 | 4.245 | 1976 | 3.980 | 0.938 |
| | 300-400 | 42392 | 3.682 | 1661 | 3.346 | 0.909 |
| | 400-500 | 37545 | 3.261 | 1825 | 3.676 | 1.127 |
| | 500-600 | 33854 | 2.940 | 1783 | 3.592 | 1.222 |
| | 600-700 | 34362 | 2.984 | 1572 | 3.167 | 1.061 |
| | 700-800 | 33376 | 2.899 | 1265 | 2.548 | 0.879 |
| | 800-900 | 32301 | 2.805 | 1069 | 2.153 | 0.768 |
| | >900 | 756037 | 65.660 | 33910 | 68.306 | 1.040 |


Therefore, the greater the ratio above unity, the stronger the relationship between
landslide occurrence and the given factor's class attribute, and the lower the ratio below
unity, the lesser the relationship between landslide occurrence and the given factor's
class attribute (Lee and Pradhan, 2006; Yalcin et al., 2011). To calculate the landslide
susceptibility index (LSI), each factor's frequency ratio values were summed as shown
in Equation 9.
$$LSI = \sum_{i=1}^{n} FR \tag{9}$$

The LSI map was reclassified using the equal interval method in GIS, and as a result,
the study area was divided into five susceptibility classes: very low, low, moderate high,
and very high (Fig. 5). According to this landslide susceptibility map, 24.67 % of the
total area was determined to be very low susceptible. Low, moderate, and high
susceptible zones constitute 23.08 %, 19.70 %, and 16.85 % of the area, respectively.
The very high susceptible area is 15.69 % of the total area.

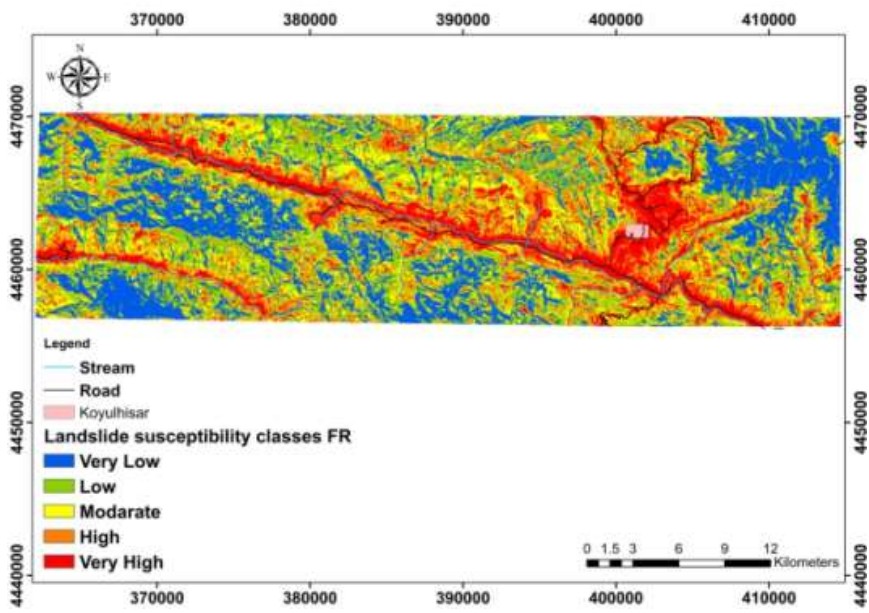


**Figure 5.** Landslide susceptibility map FR


**5.    Validation of Landslide Susceptibility map**
Landslide susceptibility maps without validation are less meaningful (Chung and Fabbri
1998). In the current study, validation of the landslides susceptibility maps was checked
by using receiver operating characteristics (ROC) (Akgun et al., 2012; Tien Bui et al.,
2012a, b, 2013; Regmi et al., 2014; Ozdemir and Altural, 2013). The ROC curve is a
useful method for representing the quality of deterministic and probabilistic detection
and forecast systems. The ROC plots the different accuracy values obtained against the
whole range of possible threshold values of the functions, and the ROC serves as a
global accuracy statistic for the model, regardless of a specific discriminate threshold
(Pourghasemi et al., 2012). In the ROC curve, the sensitivity of the model (the
percentage of existing landslide pixels correctly predicted by the model) is plotted
against 1-specificity (the percentage of predicted landslide pixels over the total study
area) (Mohammady et al., 2012; Jaafari et al., 2013). The area under the ROC curve
(AUC) represents the quality of the probabilistic model to reliably predict of the
occurrence or non-occurrence of landslides. A good fit model has an AUC values range
from 0.5–1, while values below 0.5 represent a random fit. The success rate results were
obtained by comparing the landslide training data with the susceptibility maps (Fig. 6).
AUC plot assessment results showed that the AUC values were 0.7571 and 0.7543 for
FR and IOE models and the training accuracy were 75.71 and 75.43 %, respectively.
From the results of the AUC evaluation, it is seen that both the success rate curve show
almost similar result. All the models employed in this study showed reasonably good
accuracy in predicting the landslide susceptibility of the study area.




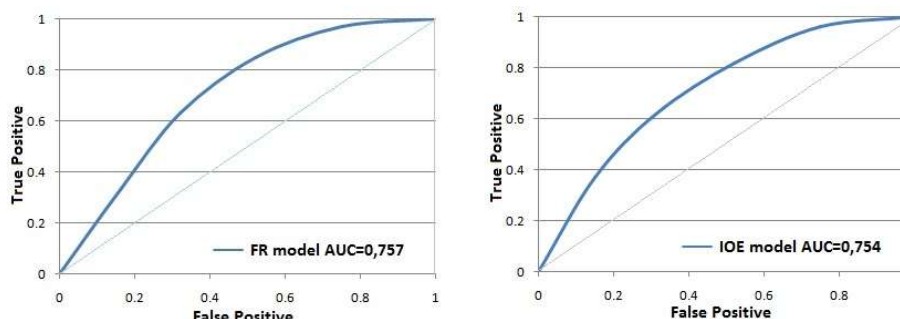

**Figure 6**. Success rate curves of FR and IOE models of the study area.

## 6. Conclusion
In this study, the accepted models, frequency ratio and index of entropy, within a GIS
environment for the aim of LSM have been used. For this purpose, six trigerring factors,
i.e., slope degree, aspect, elevation, distance to faults, distance to streams and distance
to road were used. A total of 91 landslides were identified and mapped. Out of which,
63 (69 %) were randomly selected for generating a model and the remaining 28 (31 %)
were used for validation purposes. In this study, five landslide susceptibility classes,
i.e., very low, low, moderate, high, and very high susceptibility for landsliding, were
derived with equal interval method. The validation has been determined by using the
ROC method in which the accuracy of the LS maps produced by the frequency ratio and
index of entropy models was 0.757 and 0.754, respectively for success rate technique.
This shows that all the models employed in this study showed reasonably good accuracy
in predicting the landslide susceptibility of the part of the North Anatolian Fault Zone
(Turkey). All susceptibility zones require further engineering geological and
geotechnical considerations. The increasing population pressure has forced people to
concentrate their activities on steep mountain slopes. Thus, to safeguard the life and
property from landslides, the susceptibility map can be used as basic tools in land
management and planning future construction projects in this area. The landslide
susceptibility map produced in this study can be used for optimum management by
decision makers and land use planners, and also avoidance of susceptible regions in
study area.

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
