# Peer review of "Landslide susceptibility assessment of the part of the North Anatolian Fault Zone (Turkey) by GIS-based frequency ratio and index of entropy models"

_Natural Hazards and Earth System Sciences, 2016_

## Referee Comment (RC1) · Anonymous Referee #1 · 28 Dec 2016

The manuscript deals with the landslide susceptibility assessment for the the part of the North Anatolian Fault Zone which is one of the active dextral strike-slip fault zone extending for about 1200 km along northern Turkey. The eastern part of the fault zone more prone to landslides and many destructive landslides happened in the last 20 years in the region. In the present study, 63 cases (69 %) out of 91 observed landslides were randomly selected for modeling, and the remaining 28 (31 %) cases were used for the model validation. Subsequently, landslide susceptibility maps were produced using frequency ratio and index of entropy models. The verification results show that frequency ratio model performed slightly better than index of entropy . The interpretation of the susceptibility map indicated that distance to streams, roads and

slope degree play major roles in landslide occurrence and distribution in the study area. Indeed, the area includes one of the major river system namely the Kelkit river and along which several landslides are actively occurring especially in the spring time, and the main road bounding the east to the West goes through this river system for about 200 km. The results of this this study will be useful in order to protect the regional and local natural hazards in the region and could assist planners and engineers for reorganizing and planning of future road construction in the area.

The MS stands ready for publication scientifically, but before publication I recommend the author to use a much better location map for the area (Figüre 1) indicating the location of the NAFZ and the main settlements clearly.

---

## Referee Comment (RC2) · Anonymous Referee #2 · 30 Dec 2016

The manuscript entitled "Landslide susceptibility assessment of the part of the North Anatolian Fault Zone (Turkey) by GIS-based frequency ratio and index of entropy models" submitted to the high IP NHESSD Journal, is divided into seven sub-section and focus on the analysis of a particular area

General comments

Too numerous elements are missing within this manuscript in order to reach the scientific standards allowing a publication in a high IF Journal, such as NHESSD.

The author misses precious advices from some scientific partners for (1) scientific consistency and lack of scientific innovation; (2) presenting the available data; (3) showing

and interpreting the results and (4) correct scientific English. The different points that need to be improved are detailed in the following section.

Specific comments

(1) landslide susceptibility analysis required a definition of the dependent and independent variables. The dependent variable is usually a particular landslide type that must be explained and shown by an inventory and some pictures. None of this is presented in the manuscript. In addition the author is mentioning in the abstract (line 15) the landslide-triggering factors while he is not referring to any triggers such as meteorological conditions or seismic events. Bohnhoff et al. (2016) detailed the 'Maximum Earthquake Magnitudes along Different Sections of the North Anatolian Fault Zone' meaning that it is a region prone to earthquake-triggered landslides but the author do not mentioned any of them. How to be sure that the landslide inventory is complete? A frequency-density function would help determining that. In seismically active areas, the reviewer would advise including a PGA layer as predisposing factor.

According to the error in line 15, the author can also mix "predisposing factors" and "triggering factors" what is even worse.

The reviewer also think that the innovation of this paper is not mentioned, what is its scientific novelty?

(2) The author must explain better what are the data available for the study area. In the first line, the author mentions "the part of the North Anatolian fault zone", but is this part interesting from a scientific point of view? Did the author delineate a catchment as it must be the case for a susceptibility zonation?

It is not mentioned when was the field work was done? What are the characteristics of the satellite images used? The description of the landslide inventory is also missing, is it a geomorphological inventory or an event inventory, what is the landslide type considered? What are their dimensions/extent?

Why did you consider so many classes (i.e. 10) for the "distance from road" factor? Is it really meaningful?

(3) The results are terribly badly presented, there is so much "white" free space on the figs. and they are of poor quality. Some zoom to particular areas could help understanding if the results are reliable or not. Where are the false negative are located? Is there at the same locations both models? I would suggest to cross-check both results in order to check similarities and differences. The author mentioned line 259 that " All susceptibility zones require further engineering geological and geotechnical considerations.", but there is no more information about the need of these consideration, why?

A separated table should consider Hj Hjmax Ij Wj parameters as they are not calculated for each classes.

The author mentions twice (lines 153 and 160) that Wj represents the resultant weight value for the parameter as a whole, but this definition if not clear to the reviewer.

(4) There a many typos along the manuscript and the author is not always using proper English for scientific high standard publication. However while the review's mother tongue is not English, he will not focus on this issue.

Technical corrections

Spaces missing at line 40, 41, 42, 50...107.. in figure caption 3...

Spaces to be remove at line 53

At line 60: the author assumes using only one model while he is presenting two.

Line 107: "the" should be remove as always when mentioning a reference Line 122: "aspect and elevation were also extracted.." instead of " aspect and elevation also were extracted.."

---

## Author Comment (AC1) · 30 Dec 2016

Dear Editor, I send a new location map for the area. I hope that these map is much better. Sincerely.

[Figure]

**Fig. 1.**